# Evaluation of the Wound Healing Potential of *Hypericum triquetrifolium* Turra: An Experimental Animal Study and Histopathological Examination

Tamam El-Elimat [1,*], Haya S. El-Qaderi [1], Wael M. Hananeh [2], Mahmoud M. Abu AlSamen [1], Ahmed H. Al Sharie [3], Musa A. Alshehabat [4], Mohammad Al-Gharaibeh [5] and Feras Q. Alali [6,*]

[1] Department of Medicinal Chemistry and Pharmacognosy, Faculty of Pharmacy, Jordan University of Science and Technology, Irbid 22110, Jordan

[2] Department of Veterinary Pathology and Public Health, Faculty of Veterinary Medicine, Jordan University of Science and Technology, Irbid 22110, Jordan

[3] Faculty of Medicine, Jordan University of Science and Technology, Irbid 22110, Jordan

[4] Department of Clinical Veterinary Medical Sciences, Faculty of Veterinary Medicine, Jordan University of Science and Technology, Irbid 22110, Jordan

[5] Department of Plant Production, Faculty of Agriculture, Jordan University of Science and Technology, Irbid 22110, Jordan

[6] College of Pharmacy, QU Health, Qatar University, Doha 2713, Qatar

\* Correspondence: telimat@just.edu.jo (T.E.-E.); feras.alali@qu.edu.qa (F.Q.A.)

**Abstract:** The wound healing potential of the aerial parts of *Hypericum triquetrifolium* Turra (Hypericaceae) was evaluated using an in vivo excision wound model in rats. Adult male Sprague Dawley rats were randomly assigned into seven groups; blank vehicles (olive oil and petroleum jelly), negative control, treatments [*H. triquetrifolium* ethanolic extract in petroleum jelly (5% and 10%) and *H. triquetrifolium* olive oil macerate (100 and 200 g/L)], and positive control (MEBO). Treatments were applied topically once daily until the wounds had completely healed. Wound areas and contraction rates were calculated, and full-thickness samples of the healed skin were collected for histopathological examination. *H. triquetrifolium* ointment (5%) showed the best wound healing activity with statistically significant differences when compared with the MEBO, petroleum jelly, and the negative control groups. Tissue sections were histopathologically examined in terms of re-epithelialization, granulation tissue development, collagen deposition, inflammatory cell infiltration, angiogenesis, and ulcer formation to support the in vivo excision wound model findings. *H. triquetrifolium* ointment (5%) showed the best histopathological scores in both re-epithelialization and ulcer formation. For quality control purposes, a high-performance liquid chromatography (HPLC) method was used to quantify key marker compounds in the extract, namely hypericin and rutin which showed a content of 0.64% and 4.46% (*w*/*w*), respectively. Based on the experimental results, *H. triquetrifolium* ointment (5%) exhibits remarkable wound healing properties at various stages of the wound healing process. Further investigations to prove its safety and efficacy in different types of wounds and to uncover its cellular mechanisms are warranted.

**Keywords:** wound healing; excision wound; *Hypericum triquetrifolium*; histopathology; olive oil; hypericin

## 1. Introduction

Skin is the largest organ in the body, accounting for about 15% of the total adult body weight [1]. As the body's outermost layer, it protects from toxins and serves a number of essential functions [2]. Skin damage triggers a natural physiological multistep restoration process that heals the wound partially or completely [3]. Wound healing involves three overlapping phases: inflammation, proliferation, and remodeling [4]. During the inflammatory phase, hemostasis, chemotaxis, and increased vascular permeability prevent

further tissue damage, close the wound, remove cellular debris, and stimulate cellular migration [4]. The proliferation phase is characterized by granulation tissue formation, re-epithelialization, and neovascularization [4]. During the remodeling phase, the wound reaches its maximum strength [4].

The healing process can be affected by a variety of factors, including improper diet, infection, hypoxia, drugs, aging, and diabetes [5]. Thus, wound healing products are being used to maintain an optimum healing environment for a complete healing process [6] Several pharmaceutical products derived from natural sources have been shown to promote wound healing, such as Solcoseryl (calf blood extract) [7,8], MEBO (sesame oil, beeswax, *β*-sitosterol) [9], and Acheflan (*Cordia verbenacea*) [10].

*Hypericum perforatum* (St. John's wort) is a well-known medicinal plant that has been used for a long time as a traditional dermatological agent for the treatment of minor wounds, cuts, burns, abrasions, bruises, and ulcers. Its activity has been supported by both in vitro and in vivo studies [11]. A closely related species that could be potentially used for wound healing is *Hypericum triquetrifolium* Turra (Hypericaceae). *H. triquetrifolium* is commonly referred to as wavy-leaf St. John's wort, curled leaf St. John's wort, tangled Hypericum; Dirnah (Lebanon), and locally known in Jordan as "Roja". It is a perennial herb, that grows up to 50 cm high with a dense tangle of thin branches, smooth but dotted with small black glands. The leaves are opposite and simple, 5–15 mm long, with wavy ends. Flowers are stalked in groups of 2–5 at the branches end, with five free yellow petals [12].

*H. triquetrifolium* is used traditionally as a sedative, astringent, and anti-spasmodic, for intestine and bile disorders, and treatment of burns [13]. Anti-inflammatory [14,15], antioxidant [16–18], cytotoxic [19], and antinociceptive activities [20] were also reported. Chemically, the secondary metabolites identified from *H. triquetrifolium* have been classified into three main groups, namely phloroglucinols (hyperfirin and adhyperfirin); naphtho-dianthrones (hypericin, pseudo-hypericin, proto-hypericin, and protopseudo-hypericin); and flavonoids (rutin) [21]. Although *H. triquetrifolium* could be of value in the treatment of burns and wounds, its effectiveness had never been demonstrated experimentally. Herein, we present an in vivo investigation of the wound healing capabilities of *H. triquetrifolium* beside quantitative analysis of key secondary metabolites in its dried ethanolic extract.

## 2. Materials and Methods

### 2.1. Plant Material

About 1 kg of *H. triquetrifolium* was collected during the flowering stage in July 2015 from the campus of Jordan University of Science and Technology, Irbid, Jordan (32.4950° N, 35.9912° E, elevation 580 m). The collected plant material was identified by Dr. Mohammad Al-Gharaibeh, a plant taxonomist at the Faculty of Agriculture, JUST. A voucher specimen (PHS-120) was deposited at the herbarium of the Faculty of Pharmacy, JUST. The aerial parts of *H. triquetrifolium* were air-dried in shadow away from direct sunlight in a well-ventilated area. The dried plant material was ground to powder using an electrical laboratory mill, stored at room temperature, and protected from light until further use.

### 2.2. General Experimental Procedures

High performance liquid chromatography (HPLC) was performed using an Agilent technologies 1200 series instrument (Agilent Technologies, Waldbronn, Germany) equipped with quaternary pump G1311A, autosampler G1392A, autosampler Thermostat G1330B, thermostated column compartment (TCC) G1316A, and UV-Vis detector (DAD) G1315D. Data processing was performed on Agilent ChemStation Software 4.2 (Agilent Technologies, Waldbronn, Germany). Samples were dried using CH-9230 rotary evaporator I-100 from BÜCHI labortechnik AG (Flawi, Switzerland) that was hooked up to a vacuum pump with interface I-100, recirculating chiller F-100 and heating bath B-100. Samples were sonicated using an Elmasonic S30/(H) VC505 ultrasonic cleaning unit (Elma electronics, Stuttgart, Germany). Sample vortexing was performed using FINEPCR Bio-active Fine Vortex (Keumjeong-dong, Republic of Korea). Samples were weighed using a Citizen

CX265N Semi Micro analytical balance (New Delhi, India), and pH was measured using an Elmetron CP-505 pH meter (Witosa, Poland). Biphenyl column (Pinnacle II PAH) from Restek was used for hypericin analysis (4 μm, 150 × 3.2 mm), whereas for rutin, Eclipse XDB-$C_{18}$ (5 μm, 150 × 4.6 mm) column was used.

Standard compounds used for HPLC were rutin 97% from ACROS organics (Thermo Fisher Scientific, Waltham, MA, USA) and hypericin 95% from Fluka Chemie (Sigma-Aldrich, St. Louis, MO, USA). Chemicals used were acetonitrile 99.99% (Fisher Chemical, Waltham, MA, USA), ethyl acetate 99.98% (Fisher Chemical), methanol 99.9% (Anaqua Chemicals, Wilmington, NC, USA), formic acid 98% (Gainland Chemical Co., Deeside, UK), sodium phosphate dibasic dehydrate 99.5–101% (Sigma Aldrich) and phosphoric acid 98% (Avonchem Ltd., Cheshire, UK).

Preparing the tissues for histopathological examination was performed by spin tissue processor Microm STP 120 from Thermo Fisher Scientific (Walldorf, Germany). After dehydration, the tissues were embedded by paraffin embedding system from Medax (Neuberg, Germany). High performance microtome system Microm HM 315 was used for sectioning the paraffin-embedded specimens with MX35 Ultra microtome blade 34°/80 mm, both from Thermo Fisher Scientific (Walldorf, Germany). Chemicals used for histopathology were xylene 98.5% (CARLO ERBA Reagents GmbH, Emmendingen, Germany), ethanol absolute AR grade 99.7% (Fischer Chemicals AG, Zürich, Switzerland), Harris haemtoxylin nuclear stain and Eosin Y histological stain (Leica Biosystems, Wetzlar, Germany), Masson's trichrome stain (Sigma-Aldrich), and formaldehyde 37% (Fischer Chemicals). For extraction, ethanol and methanol used were 95% (analytical grade), while hexane was 99% (HPLC grade).

## 2.3. Extraction and Formulation

Olive oil macerate was prepared by placing about 100 and 200 g of dried and powdered aerial parts of *H. triquetrifolium* in two separate transparent glass jars containing about 1000 mL of olive oil. The jars were left under sunlight daily for 4 weeks. After that, the oil was filtered and used for the subsequent experiments without any further processing as previously described [22].

Using 95% ethanol, 500 g of finely ground aerial parts of dried *H. triquetrifolium* were extracted extensively using a Soxhlet apparatus at 65 °C. The solvent was filtered and evaporated under reduced pressure using a rotary evaporator at 40 °C. The dried extract (107.3 g; 21.5% *w/w*) was then reconstituted in a 400 mL mixture of 9:1 MeOH:$H_2O$ and transferred to a separatory funnel. Hexane (800 mL × 3) was added to the aqueous methanolic extract. The biphasic solution was shaken vigorously and then the hexane layer was drawn off. The aqueous methanolic layer was evaporated to dryness under reduced pressure using a rotary evaporator at 40 °C to yield ~51 g (10.2% *w/w*) of *H. triquetrifolium* extract.

Two concentration levels (5% and 10%) of *H. triquetrifolium* extract were prepared by mixing over a glass plate using 5 and 10 g of the dried defatted ethanolic extracts with 95 and 90 g of petroleum jelly, respectively. The prepared ointments were then transferred into appropriate jars and stored in a refrigerator until required for the animal study.

## 2.4. High-Performance Liquid Chromatography (HPLC) Analysis

The ethanolic extract of *H. triquetrifolium* was analyzed using HPLC for quantitative analysis of specific marker compounds, namely hypericin and rutin. Two stock solutions of *H. triquetrifolium* ethanolic extract (10,000 and 20,800 ppm) were prepared for rutin and hypericin quantification, respectively. Stock solutions of both hypericin and rutin authentic standards were used to prepare five calibration-curve points and two quality control (QC) samples. For rutin, the calibration-curve points were 7.81, 15.63, 62.5, 125, and 500 ppm, whereas the QC points were 31.25 and 250 ppm. For hypericin, the calibration-curve points were 1.56, 3.13, 6.25, 25, and 100 ppm, and the QC points were 12.5 and 50 ppm.

An isocratic mobile phase consisting of a mixture of methanol: phosphate buffer: ethyl acetate (360:117.5:100) with a pH of 2.1 ± 0.1 was used for hypericin analysis. The flow rate used was 1 mL/min, the detector was set at 590 nm, and the injection volume was 40 μL. The total run time was 10 min [23]. For rutin, a gradient mobile phase consisting of water acidified with 0.1% formic acid (A) and acetonitrile (B) was used. The gradient used in the analysis was 5% (B) for 0–2 min, 5–90% (B) for 2–17 min, 90% (B) for 17–23 min and 90–5% (B) for 23–30 min. The flow rate used was 1 mL/min, the detector was set at 210 nm, 254 nm, and 280 nm, and the injection volume was 30 μL. The total run time was 30 min [24,25]. The plant samples along with the calibration and QC were injected in triplicate.

### 2.5. Animals and Treatments

Adult male Sprague Dawley rats weighing ~220–240 g were obtained from the animal breeding facility at Jordan University of Science and Technology (JUST) (Irbid, Jordan). Rats were kept in small plastic cages (one rat per cage) under optimum hygienic conditions. The animals were left for 3 days at room temperature for acclimatization. They were housed in an air-conditioned room at a temperature of 24 °C with standard pellet feed and water ad libitum. Rats were identified by tagging their tails and kept under typical lighting conditions with a 12-h cycle. The study procedure was approved by the Animal Care and Use Committee (ACUC) of the Jordan University of Science and Technology. All procedures were conducted per the National Research Council's Guide for the Care and Use of Laboratory Animals.

Wound healing was evaluated using an in vivo excision wound model in rats (Figure 1). Rats were randomly assigned to eight groups (*n* = 8), namely blank vehicles, negative control, treatments, and positive control. The animals in the negative control group (G1) were not treated with any product. While the animals in the blank vehicle groups were topically treated with either an ointment base (G2) consisting of petroleum jelly (Vaseline BLUSEAL) or olive oil (G3). The animals in the positive control group (G4) were treated with MEBO. Animals in the treatment groups were treated with *H. triquetrifolium* [5% ointment (G5) and 10% ointment (G6)], and olive oil macerate of *H. triquetrifolium* [100 g/L (G7) and 200 g/L (G8)].

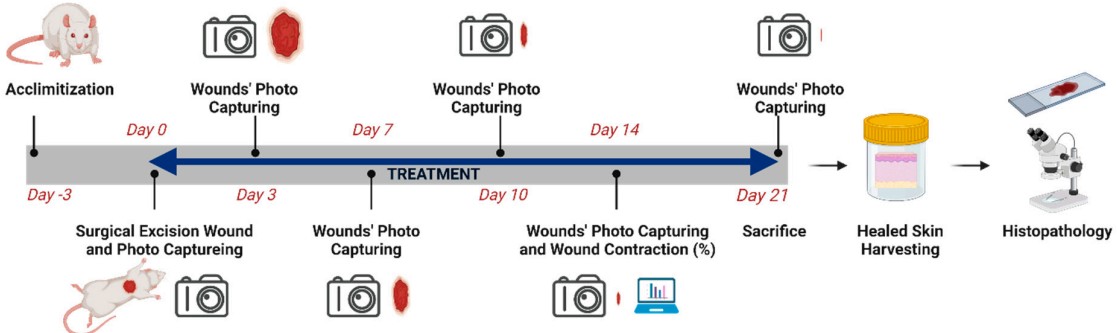

**Figure 1.** Schematic illustration of the experimental design.

### 2.6. Excision Wound Model

An excision wound model was used to monitor wound contraction and wound closure time. Animals in each group were anesthetized, and the back hair of the rats was depilated using an electrical shaving machine. The wounds (2 × 2 cm; 400 mm²) were created on the dorsal region of each animal by excising the skin surgically. The animals were weighed every 3–4 days. The wounds were then cleaned with 10% iodine and left open without any dressing. The tested extracts, reference drugs, and vehicles were applied topically once daily until the wounds had completely healed. The progressive changes in the wound area were monitored by capturing photos using a Canon ELPH 100 HS camera (Canon, Tokyo, Japan) on the following days: 0, 3rd, 7th, 10th, 14th and 21st. Using ImageJ software

(US National Institutes of Health (NIH), Bethesda, MD, USA), and the wound area was calculated. Wound contraction was calculated as the percentage of the reduction in the wounded area using the following equation:

$$\text{Wound contraction percentage} = \frac{\text{WA}(0) - \text{WA}(\text{x})}{\text{WA}(0)} \times 100\%$$

where WA(0) represents the wound area at time zero and WA(x) represents the wound area at time x. Following the experiment, full-thickness skin samples from the intersection area between the healed and normal skin of all rats were collected for histopathological examination.

### 2.7. Histopathology

The skin tissue was prepared for histopathological examination as described by Avti, P.K., et al. [26]. After harvesting, the tissues were fixed in 10% formaldehyde for 24 h. The tissues were then dehydrated in ascending grades of alcohol and cleared with xylene. Dehydration was achieved by passing tissues through a graded series of alcohol followed by two changes of xylene. After infiltration in paraffin wax, tissues were embedded in pure paraffin wax [26]. Thin sections of about 5 μm in thickness were obtained by microtome. Finally, sections were mounted on glass slides and stained with hematoxylin and eosin (H&E) stain. Some featured slides were stained with Masson's trichrome special stain. Sections were examined using a compound light microscope (B-380 Series, OPTIKA, Italy). Skin tissues were scored histopathologically in terms of re-epithelialization, granulation development, collagen accumulation, inflammatory cell infiltration, angiogenesis, and ulcer formation [27].

### 2.8. Statistical Analysis

The wound area data were analyzed using IBM SPSS Statistics for Windows, version 25 (IBM Corp., Armonk, NY, USA). In terms of descriptive analyses, wound area size, and wound contraction at days 3, 7, 10, and 14 as compared to baseline at day 0 were described as mean ± standard error of the mean (SEM). For visualization purposes, the percentage of unhealed wound area was computed from (100—wound contraction). In terms of inferential analysis, the data were tested for normality using the Kolmogorov-Smirnov test but failed to meet normality requirements. Therefore, parametric analyses were not used and the non-parametric Kruskal-Wallis H test was used instead to analyze the median unhealed wound area on day 14 compared to baseline followed by Bonferroni pairwise correction for multiple comparisons. A $p$-value $\leq 0.05$ was considered statistically significant. All graphs were generated using GraphPad Prism version 7.00 for Windows (GraphPad Software, La Jolla, CA, USA).

## 3. Results and Discussion

The wound healing potential of the ethanolic extract and olive oil macerate of *H. triquetrifolium* was evaluated using an in vivo excision wound model in rats as well as histopathological examination. Wound areas were recorded on days 0, 3rd, 7th, 10th, 14th, and 21st (Table 1) with their corresponding representative photos (Figure 2A). The animals were closely monitored for any symptoms of local or systemic reactions. The wounds were checked routinely for any signs of inflammation and/or purulent exudate formation. A purulent exudate was observed in four of the animals, so they were excluded from the study. Additionally, animal mortalities were recorded. Three animals died throughout the experiment.

**Table 1.** Effects of the investigated formulations on excision wound model.

| Treatment | Mean Wound Area (cm$^2$) $\pm$ S.E.M (% Contraction Rate) at Specific Days | | | | |
|---|---|---|---|---|---|
| | 0 | 3 | 7 | 10 | 14 |
| Negative control (G1) | 3.75 $\pm$ 0.20 | 3.1 $\pm$ 0.27 (16.90) | 1.34 $\pm$ 0.11 (63.88) | 0.54 $\pm$ 0.08 (85.63) | 0.19 $\pm$ 0.04 (94.91) |
| Petroleum jelly (G2) | 3.86 $\pm$ 0.16 | 3.08 $\pm$ 0.21 (19.79) | 2.76 $\pm$ 0.73 (27.26) [a] | 1.07 $\pm$ 0.52 (72.93) [a] | 0.48 $\pm$ 0.15 (87.57) |
| Olive oil (G3) | 3.92 $\pm$ 0.15 | 2.31 $\pm$ 0.23 (40.31) | 1.55 $\pm$ 0.80 (60.84) [b] | 0.60 $\pm$ 0.10 (83.94) | 0.29 $\pm$ 0.06 (92.71) |
| MEBO (G4) | 3.74 $\pm$ 0.18 | 2.50 $\pm$ 0.27 (33.57) | * | * | 0.21 $\pm$ 0.06 (94.18) |
| 5% *H. triquetrifolium* ointment (G5) | 3.88 $\pm$ 0.12 | 3.15 $\pm$ 0.20 (18.71) | 1.88 $\pm$ 0.04 (54.30) [a] | 0.30 $\pm$ 0.01 (91.84) | 0.08 $\pm$ 0.06 (98.10) |
| 10% *H. triquetrifolium* ointment (G6) | 3.76 $\pm$ 0.26 | 2.92 $\pm$ 0.20 (19.89) | 1.43 $\pm$ 0.23 (56.13) [a] | 0.33 $\pm$ 0.04 (90.87) | 0.11 $\pm$ 0.03 (96.86) |
| 100 g/L *H. triquetrifolium* olive oil macerate (G7) | 3.73 $\pm$ 0.13 | 1.75 $\pm$ 0.11 (52.86) | 0.95 $\pm$ 0.09 (74.07) | 0.40 $\pm$ 0.10 (89.22) | 0.17 $\pm$ 0.03 (95.46) |
| 200 g/L *H. triquetrifolium* olive oil macerate (G8) | 4.10 $\pm$ 0.21 | 1.93 $\pm$ 0.15 (50.37) | 1.02 $\pm$ 0.72 (71.68) [b] | 0.45 $\pm$ 0.15 (87.77) | 0.20 $\pm$ 0.04 (94.93) |

* Data are missing due to presence of scab that renders the measurement of the wound area difficult. [a] Statistics were done for only 3 data points (due to the presence of scab); [b] Statistics were done for only 2 data points (due to the presence of scab).

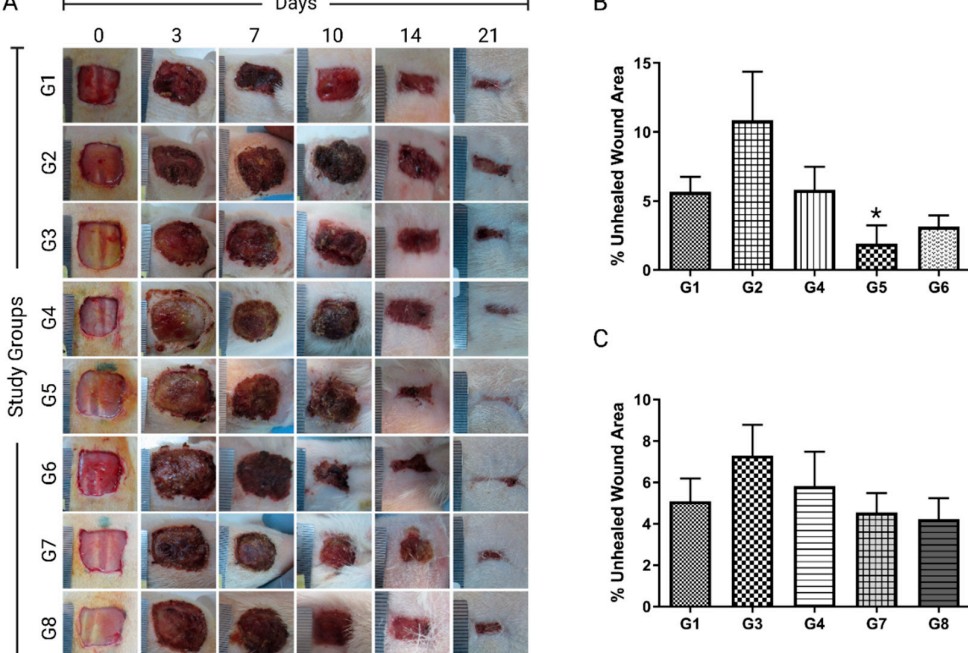

**Figure 2.** Photographs of wounds and their respective unhealed wound area percentage by different treatment groups (**A**). Negative control (G1), petroleum jelly (G2), olive oil (G3), MEBO (G4), 5% *H. triquetrifolium* ointment (G5), 10% *H. triquetrifolium* ointment (G6), 100 mg/L *H. triquetrifolium* olive oil macerate (G7), and 200 mg/L *H. triquetrifolium* olive oil macerate (G8). Wound photographs scales are represented in millimeters. Unhealed wound area across test groups (**B**,**C**). * Denotes for statistical significance ($p \leq 0.05$) compared to the corresponding vehicle.

The ointment formulation of *H. triquetrifolium* in two concentrations (5 and 10%) was tested for statistical differences against the negative control, petroleum jelly, and MEBO groups. The Kruskal-Wallis H test showed that the median percentage unhealed wound area was significantly different among different treatment groups $\chi^2(4) = 11.86$, $p = 0.018$. A pairwise comparison showed that 5% *H. triquetrifolium* demonstrated a wound healing potential when compared to negative control ($p = 0.008$), MEBO ($p < 0.019$), and petroleum jelly ($p < 0.008$) (Figure 2B). The wound contractions were 91.84% and 98.10% for the 5% *H. triquetrifolium* ointment treated group on days 10 and 14, respectively (Table 1 and Figure 2B). In summary, the topical application of 5% *H. triquetrifolium* extract ointment onto the excised model demonstrated the best results with the lowest wound area (98.10%) on day 14. It significantly reduced the rate of wound healing compared to no-treatment,

petroleum jelly, or MEBO-treated groups (Table 1 and Figure 2B). A similar study was conducted on the ethanolic extract of *H. perforatum* (St. John's Wort) [28]. Eucerin base at 2, 5, and 10% *w/w* were tested on full-thickness excision wound in rabbits. Phenytoin cream (1%) was used as a standard healing agent. Compared to the negative control, Eucerin, and phenytoin; the *H. perforatum* ethanolic extract significantly reduced the rate of wound healing. The best concentration among all was the 2% *H. perforatum* cream ($p < 0.01$) [28].

In the current study, the olive oil macerate of *H. triquetrifolium* had not shown any significant differences in wound healing activity compared to the negative control, olive oil, or MEBO groups (Table 1 and Figure 2C). However, an olive oil macerate of *H. perforatum* had shown a significant wound healing effect on the excision wound model (Süntar et al., 2010). When testing the wound healing activity of *H. triquetrifolium* macerate, the Kruskal-Wallis H test showed no statistically significant difference in the median percentage unhealed wound area between the different treatment groups (negative control, olive oil, MEBO, 100 g/L *H. triquetrifolium* and 200 g/L *H. triquetrifolium*/olive oil), $\chi^2(5) = 2.143$, $p = 0.82$ (Table 1 and Figure 2C).

The healed tissues were scored histopathologically (Supplementary Table S1) in terms of re-epithelialization, granulation development, collagen deposition, inflammatory cell infiltration, angiogenesis, and ulcer formation using a qualitative scale. Histopathological evaluation for each group was done at the end of the experiment (Table 2). Histopathological results were in agreement with the photographs of some featured tissues, which were stained with H&E stain and Masson's trichrome special stain (Figure 3 and Figures S1–S8).

**Table 2.** Histopathological evaluation of wound healing processes and phases treated with different investigated formulations.

| Treatment | Re-Epithelialization | Granulation Tissue | Collagen Accumulation | Inflammatory Cells | Angiogenesis | Ulcer |
|---|---|---|---|---|---|---|
| Negative control (G1) | $1.17 \pm 0.31$ | $2.33 \pm 0.21$ | $2.33 \pm 0.21$ | $1.83 \pm 0.60$ | $2.67 \pm 0.21$ | $2.17 \pm 0.54$ |
| Petroleum jelly (G2) | $0.60 \pm 0.22$ | $1.60 \pm 0.22$ | $1.60 \pm 0.22$ | $2.00 \pm 0.41$ | $3.00 \pm 0.00$ | $2.80 \pm 0.18$ |
| Olive oil (G3) | $1.00 \pm 0.46$ | $2.75 \pm 0.16$ | $2.75 \pm 0.16$ | $1.50 \pm 0.42$ | $2.63 \pm 0.18$ | $2.13 \pm 0.48$ |
| MEBO (G4) | $1.43 \pm 0.43$ | $2.14 \pm 0.34$ | $2.14 \pm 0.34$ | $1.43 \pm 0.48$ | $2.00 \pm 0.31$ | |
| 5% *H. triquetrifolium* ointment (G5) | $2.13 \pm 0.44$ | $2.00 \pm 0.33$ | $2.25 \pm 0.31$ | $1.63 \pm 0.46$ | $2.25 \pm 0.37$ | $1.00 \pm 0.50$ |
| 10% *H. triquetrifolium* ointment (G6) | $1.67 \pm 0.33$ | $2.33 \pm 0.33$ | $2.33 \pm 0.33$ | $1.50 \pm 0.56$ | $2.50 \pm 0.34$ | $1.17 \pm 0.40$ |
| 100 g/L *H. triquetrifolium* olive oil macerate (G7) | $1.50 \pm 0.38$ | $1.88 \pm 0.23$ | $1.88 \pm 0.23$ | $2.25 \pm 0.41$ | $2.13 \pm 0.30$ | $1.88 \pm 0.48$ |
| 200 g/L *H. triquetrifolium* olive oil macerate (G8) | $1.75 \pm 0.37$ | $2.50 \pm 0.19$ | $2.50 \pm 0.19$ | $2.00 \pm 0.42$ | $2.75 \pm 0.16$ | $1.88 \pm 0.55$ |

All data is described as mean ± SEM. Statistical analysis did not show any statical significance among test groups ($p > 0.05$).

The normal skin is composed of a continuous layer of stratified squamous epithelium that is called the epidermis. This layer overlies a dermis layer that is composed of connective tissues and encompasses hair follicles, sweat glands, blood vessels, and other structures (Supplementary Figure S1).

Within the examined skin sections, different animals failed either completely or partially to close the wound area leaving the dermis exposed to scab formation (Supplementary Figure S2). Other animals exhibited complete wound re-epithelialization where a continuous layer of stratified epithelium covered the entire length of the wound area (Supplementary Figure S3). Several degrees of fibroplasia were observed in the dermis visualized by the Masson trichrome stain (Supplementary Figures S4 and S5). Mature connective tissue formation with perpendicular new capillary formation was also seen (Supplementary Figure S4). Variable degrees of angiogenesis were present (Supplementary Figure S6). Deep inflammatory cell aggregates were seen (Supplementary Figure S7). In comparison with the normal dermis (Supplementary Figure S8b), in some sections, the dermis of the completely re-epithelialized skin wound was characterized by the lack or

the presence of few inflammatory cells, presence of fibroplasia, and lack of hair follicles, sebaceous glands, or sweat glands (Supplementary Figure S8a).

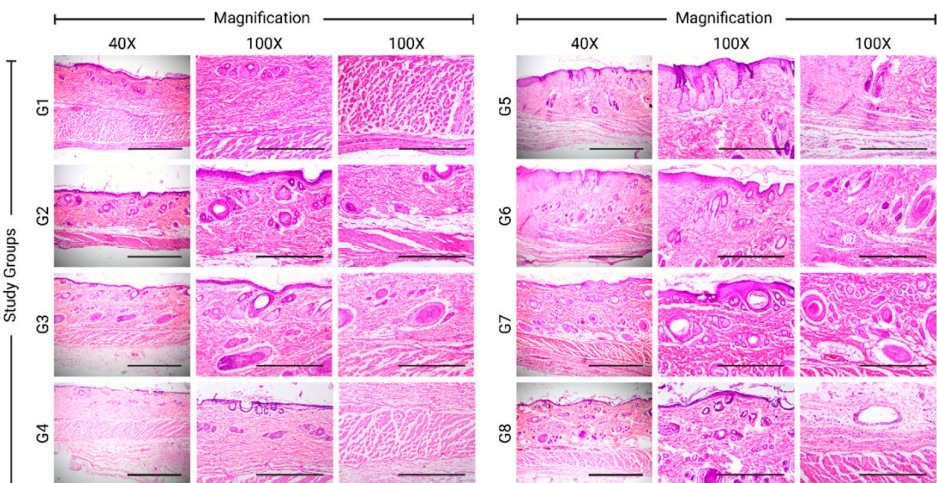

**Figure 3.** Histopathological evaluation of healed tissues by different treatment groups stained with hematoxylin and eosin (H&E) on day 21. Negative control (G1), petroleum jelly (G2), olive oil (G3), MEBO (G4), 5% *H. triquetrifolium* ointment (G5), 10% *H. triquetrifolium* ointment (G6), 100 mg/L *H. triquetrifolium* olive oil macerate (G7), and 200 mg/L *H. triquetrifolium* olive oil macerate (G8). 40× scale is 1 mm and 100× scale is 500 μm.

In the experimental groups, the 5% *H. triquetrifolium* ointment group scored the highest reepithelization compared to the negative control, petroleum jelly, and the MEBO groups, whereas the 10% *H. triquetrifolium* ointment illustrated a weaker reepithelization. The olive oil and petroleum jelly groups seem to worsen the process of reepithelization. When the *H. triquetrifolium* plant material was incorporated with olive oil as a macerate formulation with a concentration of 100 and 200 g/L, it induced a moderate reepithelization. However, the olive oil group induced the highest deposition of granulation tissue compared to the negative control group. The 200 g/L *H. triquetrifolium* oil macerate scored high in terms of granulation deposition compared to the 100 g/L *H. triquetrifolium* oil macerate. The same pattern was also observed in the ointment's formulations in which the 10% *H. triquetrifolium* ointment has better deposition of connective tissue compared to the 5% *H. triquetrifolium* ointment.

The number of inflammatory cells seen within the histopathological examination of 5% and 10% *H. triquetrifolium* ointment groups is proportionally less compared to the negative control group. The macerate formulations were scored with a high number of inflammatory mediators when testing the 100 and 200 g/L *H. triquetrifolium*. Angiogenesis was graded based on the number of veins observed in the H&E-stained histopathological slides. The highest angiogenesis score was observed in the petroleum jelly group compared to the negative control group. The positive control group (MEBO) scored low in terms of wound vascularization even when compared to the negative control and the olive oil groups. Only the 200 g/L *H. triquetrifolium* macerate group scored a higher level of vascularization compared to the negative control. Other tested *H. triquetrifolium* formulations worsened the angiogenesis process with a lower histopathological score compared to the negative control.

Regrading ulcer formation, the 5% and the 10% *H. triquetrifolium* ointment groups were remarkably given lower scores for the occurrence of skin ulceration when compared to the negative control. In addition, the 100 and 200 g/L *H. triquetrifolium* macerate groups also showed a low occurrence of epithelial ulceration compared to the negative control. The petroleum jelly group showed higher ulceration as opposed to the positive control group (MEBO) which showed a lower incidence of ulceration.

In summary, the histopathological analysis showed that the 5% *H. triquetrifolium* ointment formulation demonstrated the best remodeling, particularly re-epithelialization, and the least ulceration.

For quality control purposes, the ethanolic extract of *H. triquetrifolium* was subjected to HPLC analysis to quantify specific marker compounds namely rutin and hypericin. A linear calibration curve for each standard compound was constructed (Supplementary Figure S9); the coefficient of determination ($r^2$) and the range for each curve are given in Table S2 (Supplementary Materials). *H. triquetrifolium* ethanolic extract was found to contain 0.64% ($w/w$) of hypericin, whereas rutin content was 4.46% ($w/w$) (Supplementary Figure S10).

In the current work, *H. triquetrifolium* extract ointments and oil macerate formulations were tested for their wound healing potential using an in vivo excision wound model in adult male Sprague Dawley rats. To the best of our knowledge, the current study describes, for the first time, the wound-healing capabilities of *H. triquetrifolium*. The 5% *H. triquetrifolium* extract ointment demonstrated potent wound-healing activity. It was higher than that of the reference drug MEBO. Future work should be guided to evaluate the effect of *H. triquetrifolium* extract in several modified wound healing models as in the case of infected wounds, diabetic wounds, or burns. The exact molecular mechanism should be also investigated along with the short- and long-term toxicological effects of *H. triquetrifolium*.

**Supplementary Materials:** The following supporting information can be downloaded at: https://www.mdpi.com/article/10.3390/scipharm91010016/s1, Figure S1: Skin, excisional wound, rat. Normal skin. Dermis is intact and adnexal structures are present (hair follicle, sweat glands). H&E stain. 4× and 10×; Figure S2: Skin, excisional wound, rat. Incomplete wound healing. Ulcer score 3, pinkish colour indicates serum. Crust formation (scab). H&E stain. 10×; Figure S3: Skin, excisional wound, rat. Complete re-epithelialization. The dermis is markedly hypercellular by fibroplasia and collagen deposition, new blood vessel formation that were perpendicular to the fibroblast at day 21st. H&E stain. 10×; Figure S4: Skin, excisional wound, rat. The skin tissue is completely healed with few inflammatory cells and fibroplasia. The epidermis is one layer and continuous at day 21st. H&E stain. 10×; Figure S5: Skin, excisional wound, rat. Completely healed skin tissue. The epidermis is one and continuous at day 21st. MT stain. 10×; Figure S6: Skin, excisional wound, rat. The tissue skin with score 3 angiogenesis that has more than 10 veins in the wounded area at day 21st. H&E stain. 10×; Figure S7: Skin, excisional wound, rat. Deep inflammation; deep focal inflammatory cells and granuloma formation. H&E stain. 4× and 10×; Figure S8: Skin, excisional wound, rat. (a) Completely healed skin with marked connective tissue (b) normal un-wounded skin with marked connective tissue. (a) the wound was completely closed and covered with mature epidermal squamous cells. The skin wound sections were relatively normal except for adnexal structure that were completely absent at day 21st. Collagen accumulation is denser than normal tissue. (b) normal un-wounded skin with adnexal structures present, and less collagen accumulation. MT stain. 10×; Figure S9: Calibration curves for rutin (A) and hypericin (B); Figure S10: HPLC chromatograms for H. triquetrifolium extract (a) chromatogram at λ = 590 nm for hypericin detection and (b) chromatogram at λ = 254 nm for rutin detection; Table S1: Scoring criteria for histopathological evaluation of the wound healing process; Table S2: Calibration curves' linear regressions ($r^2$) and range.

**Author Contributions:** T.E.-E and F.Q.A. conceptualized the research idea, provided the resources, and supervised the work. H.S.E.-Q., W.M.H., M.A.-G., M.M.A.A. and M.A.A.: conducted experiments. H.S.E.-Q., W.M.H., M.M.A.A. and A.H.A.S.: analyzed the data. T.E.-E. and A.H.A.S. prepared the visuals. H.S.E.-Q. and M.A-G. wrote the original draft. The authors declare that all data were generated in-house and that no paper mill was used. All authors have read and agreed to the published version of the manuscript.

**Funding:** This work was funded by a grant from the Deanship of Research, Jordan University of Science and Technology (Grant number: 122/2017) and Qatar University.

**Institutional Review Board Statement:** The study procedure was approved by the Animal Care and Use Committee (ACUC) of the Jordan University of Science and Technology. All procedures were conducted per the National Research Council's Guide for the Care and Use of Laboratory Animals. Consent to Participate Not applicable. Consent to Publish Not applicable.

**Informed Consent Statement:** Not applicable.

**Data Availability Statement:** All data generated during this study are included in this published article and Supplementary Materials.

**Conflicts of Interest:** The authors declare no conflict of interest.

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
