# Peer review of "Evaluation of the Wound Healing Potential of Hypericum triquetrifolium Turra: An Experimental Animal Study and Histopathological Examination"

_scipharm, doi:10.3390/scipharm91010016_

Round 1
Reviewer 1 Report
The authors investigated the wound-healing effect of extracting the plant. The extract has been traditionally used, and the results obtained from the study seem to be valuable.
(1) Petroleum jelly was used in this study, but the authors describe why petroleum jelly was selected as a dispersant.
(2) The authors resulted that 5% is the best wound-healing activity. The reason should be scientifically described.
(3) Physicochemical properties of ointments should be shown and described.
Author Response
Response to reviewer #1:
Comment 1: Petroleum jelly was used in this study, but the authors describe why petroleum jelly was selected as a dispersant.
Response: The authors would thank the reviewer for highlighting this issue; petroleum jelly has been used for ages in wound healing experiments as a chemical loader. Its inert biological role alongside the chemical and physical properties renders it a perfect dispersant. It is colorless which helps in insuring a full solubility observation of the plant extract. It does not oxidize on exposure to the air and is not readily acted on by chemical reagents. In addition, it is insoluble in water which helps to maintain adhesiveness to skin insuring a full delivery of the tested extract. Finally, the semi-solid nature helps in applying it with the loaded extract into artificial wounds.
Comment 2: The authors resulted that 5% is the best wound-healing activity. The reason should be scientifically described.
Response: We believe that 5% plant extract exerts its effect through mediating multiple pathways involved in wound healing but by increasing the concentration to 10%; such modulation will be over exhibited by the toxicity of certain phytochemicals. It a pure hypothetical explanation that should be examined in future studies.
Comment 3: Physicochemical properties of ointments should be shown and described.
Response: The petroleum jelly and olive oil macerate were used as a pure dispersant to release the plant extract to the wounds. The main scope of the article is to perform a pilot study testing the biological safety and efficacy of Hypericum triquetrifolium extract in wound healing. The concept of testing several physicochemical properties including hydration properties (water activity, water absorption capacity, water retention capacity, hygroscopicity, dispersibility, solubility, etc.), rheological fluid behavior, mechanical properties, optical properties (color, translucence, etc.), and thermal properties were not implemented since we aimed to test the biological role only. Currently; the only active concentration (5% plant extract) is being under extensive studying in our lab and we are collaborating with multiple pharmaceutical specialist to compare several extract formulations in wound healing models. The same study will include a a detailed biological data which can explain comment number 2 as well.
Reviewer 2 Report
This manuscript evaluated the healing potential of Hypericum triquetrifolium Turra ointment in rats with excision wounds. The H. triquetrifolium (5%) ointment showed the best healing activity compared to negative control groups, vaseline, and positive control. The quantification of hypericin and rutin compounds was performed by HPLC, with contents of 0.64% and 4.46% (w/w), respectively. Future studies are necessary to prove their safety and effectiveness in different types of wounds and to discover their cellular mechanisms. This study has potential and should make essential minor changes for its conclusion.
Abstract:
Agreement errors: "Adult Male Sprague Dawley Rats 19 Were Randomly Assigned" should be "Adult Male Sprague Dawley Rats Were Randomly Assigned".
Agreement errors: "Wound Areas and Contraction 23 Rates Were Calculated" should be "Wound Areas and Contraction Rates Were Calculated".
Introduction:
The introduction clearly presents the problem at hand. It has the appropriate level of update and presents classic elements for the chosen approach.
Materials and Methods:
Plant material: Include the coordinates to the South, East, West, or North at the end of the latitude and longitude.
How long was it kept in the drying process? Was the constant weight mentioned?
The authors could provide better details on the preparation of the formulations used.
Excision wound model: Reduce the size of the equation.
"w/w" - Put it in italics.
Results and discussion
Figure 2: Add the metric scale to the images. In images B and C, indicate the significance of the results.
Table 2: Include the statistical significance of the results.
Figure 3: Add scales to the images.
No discussions based on references were conducted, and several points can be further explored.
Provide clearer conclusions in the discussion.
Author Response
Response to reviewer #2:
Comment 1: Abstract: Agreement errors: "Adult Male Sprague Dawley Rats 19 Were Randomly Assigned" should be "Adult Male Sprague Dawley Rats Were Randomly Assigned" And Agreement errors: "Wound Areas and Contraction 23 Rates Were Calculated" should be "Wound Areas and Contraction Rates Were Calculated".
Response: The comment has been taken into consideration; we believe that number 19 and 23 were added by mistake through the production process as they represent line numbers.
Comment 2: Introduction: The introduction clearly presents the problem at hand. It has the appropriate level of update and presents classic elements for the chosen approach.
Response: The authors would thank the reviewer for his valuable comment.
Comment 3: Materials and Methods: Plant material: Include the coordinates to the South, East, West, or North at the end of the latitude and longitude.
Response: The comment has been taken into consideration; the location of JUST campus (32.4950° N, 35.9912° E) where the plant material was collected has been added.
Comment 4: How long was it kept in the drying process? Was the constant weight mentioned?
Response: The plant material was left for around 3 weeks in shadow away from direct sunlight in a well-ventilated area. Constant weight measurement was not monitored.
Comment 5: The authors could provide better details on the preparation of the formulations used.
Response: The comment has been taken into consideration and the following part of methodology regarding the formulation was revised: Olive oil macerate was prepared by placing about 100 and 200 g of dried and powdered aerial parts of H. triquetrifolium in two separate transparent glass jars con-taining about 1000 mL of olive oil. The jars were left under sunlight daily for 4 weeks. After that, the oil was filtered and used for the subsequent experiments without any further processing as previously described (Süntar et al., 2010). Using 95% ethanol, 500 g of finely ground aerial parts of dried H. triquetrifolium were extracted extensively using a Soxhlet apparatus at 65°C. The solvent was filtered and evaporated under reduced pressure using a rotary evaporator at 40°C. The dried extract (107.3 g; 21.5% w/w) was then reconstituted in a 400 mL mixture of 9:1 MeOH:H2O and transferred to a separatory funnel. Hexane (800 mL × 3) was added to the aqueous methanolic extract. The biphasic solution was shaken vigorously and then the hexane layer was drawn off. The aqueous methanolic layer was evaporated to dry-ness under reduced pressure using a rotary evaporator at 40°C to yield ~51 g (10.2% w/w) of H. triquetrifolium extract. Two concentration levels (5% and 10%) of H. triquetrifolium extract were prepared by mixing over a glass plate using 5 and 10 g of the dried defatted ethanolic extracts with 95 and 90 g of petroleum jelly, respectively. The prepared ointments were then trans-ferred into appropriate jars and stored in a refrigerator until required for the animal study.
Comment 6: Excision wound model: Reduce the size of the equation, and "w/w" - Put it in italics.
Response: The size of the equation was reduced and "w/w" was made italic through the manuscript.
Comment 7: Results and discussion, Figure 2: Add the metric scale to the images. In images B and C, indicate the significance of the results. Table 2: Include the statistical significance of the results. Figure 3: Add scales to the images. No discussions based on references were conducted, and several points can be further explored. Provide clearer conclusions in the discussion.
Response: The comments have been taken into consideration. Figure 2 scales were clarified within the legends and statistical analysis results were presented in Figure 2 B and C as well as table 2 footnote. Histopathological images scales were added as well with their corresponding clarification within the legend. Conclusion and discussion were revised as requested.
Round 2
Reviewer 1 Report
The authors have rewritten it according to the reviewer's comments. There are no additional comments.